# Emergence of functional and structural properties of the head direction system by optimization of recurrent neural networks

**Christopher J. Cueva**[*][†], **Peter Y. Wang**,[*] **Matthew Chin**,[*] **Xue-Xin Wei**[†]
Columbia University
New York, NY 10027, USA

## Abstract

Recent work suggests goal-driven training of neural networks can be used to model neural activity in the brain. While response properties of neurons in artificial neural networks bear similarities to those in the brain, the network architectures are often constrained to be different. Here we ask if a neural network can recover both neural representations and, if the architecture is unconstrained and optimized, the anatomical properties of neural circuits. We demonstrate this in a system where the connectivity and the functional organization have been characterized, namely, the head direction circuits of the rodent and fruit fly. We trained recurrent neural networks (RNNs) to estimate head direction through integration of angular velocity. We found that the two distinct classes of neurons observed in the head direction system, the Compass neurons and the Shifter neurons, emerged naturally in artificial neural networks as a result of training. Furthermore, connectivity analysis and *in-silico* neurophysiology revealed structural and mechanistic similarities between artificial networks and the head direction system. Overall, our results show that optimization of RNNs in a goal-driven task can recapitulate the structure and function of biological circuits, suggesting that artificial neural networks can be used to study the brain at the level of both neural activity *and* anatomical organization.

## 1 Introduction

Artificial neural networks have been increasingly used to study biological neural circuits. In particular, recent work in vision demonstrated that convolutional neural networks (CNNs) trained to perform visual object classification provide state-of-the-art models that match neural responses along various stages of visual processing (Yamins et al., 2014; Khaligh-Razavi & Kriegeskorte, 2014; Yamins & DiCarlo, 2016; Cadieu et al., 2014; Güçlü & van Gerven, 2015; Kriegeskorte, 2015). Recurrent neural networks (RNNs) trained on cognitive tasks have also been used to account for neural response characteristics in various domains (Mante et al., 2013; Sussillo et al., 2015; Song et al., 2016; Cueva & Wei, 2018; Banino et al., 2018; Remington et al., 2018; Wang et al., 2018; Orhan & Ma, 2019; Yang et al., 2019). While these results provide important insights on how information is processed in neural circuits, it is unclear whether artificial neural networks have converged upon similar architectures as the brain to perform either visual or cognitive tasks. Answering this question requires understanding the functional, structural, and mechanistic properties of artificial neural networks and of relevant neural circuits.

We address these challenges using the brain's internal compass - the head direction system, a system that has accumulated substantial amounts of functional and structural data over the past few decades in rodents and fruit flies (Taube et al., 1990a;b; Turner-Evans et al., 2017; Green et al., 2017; Seelig & Jayaraman, 2015; Stone et al., 2017; Lin et al., 2013; Finkelstein et al., 2015; Wolff et al., 2015; Green & Maimon, 2018). We trained RNNs to perform a simple angular velocity (AV) integration task (Etienne & Jeffery, 2004) and asked whether the anatomical and functional features that have emerged as a result of stochastic gradient descent bear similarities to biological networks sculpted

---

[*]equal contribution
[†]Correspondence: ccueva@gmail.com, weixxpku@gmail.com

by long evolutionary time. By leveraging existing knowledge of the biological head direction (HD) systems, we demonstrate that RNNs exhibit striking similarities in both structure and function. Our results suggest that goal-driven training of artificial neural networks provide a framework to study neural systems at the level of both neural activity *and* anatomical organization.

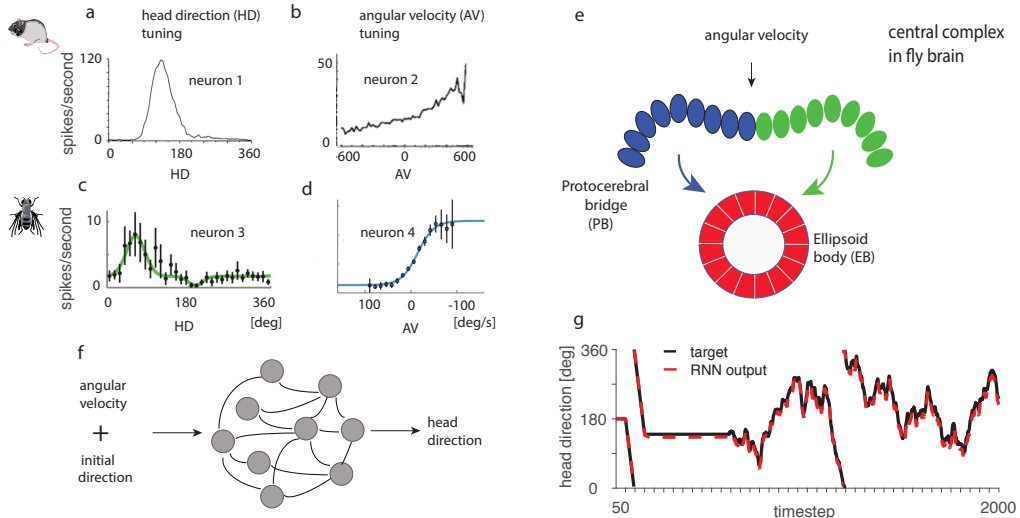

Figure 1: Overview of head direction system in rodents, fruit flies, and the RNN model. **a, c)** tuning curve of an example head direction (HD) cell in rodents (a, adapted from Taube (1995)) and fruit flies (c, adapted from Turner-Evans et al. (2017)). **b, d)** Tuning curve of an example angular velocity (AV) selective cell in rodents (b, adapted from Sharp et al. (2001)) and fruit flies (d, adapted from Turner-Evans et al. (2017)). **e)** The brain structures in the fly central complex that are crucial for maintaining and updating heading direction, including the protocerebral bridge (PB) and the ellipsoid body (EB). **f)** The RNN model. All connections within the RNN are randomly initialized. **g)** After training, the output of the RNN accurately tracks the current head direction.

## 2 MODEL

### 2.1 MODEL STRUCTURE

We trained our networks to estimate the agent's current head direction by integrating angular velocity over time (Fig. 1f). Our network model consists of a set of recurrently connected units ($N = 100$), which are initialized to be randomly connected, with no self-connections allowed during training. The dynamics of each unit in the network $r_i(t)$ is governed by the standard continuous-time RNN equation:

$$\tau \frac{dx_i(t)}{dt} = -x_i(t) + \sum_j W_{ij}^{\text{rec}} r_j(t) + \sum_k W_{ik}^{\text{in}} I_k(t) + b_i + \xi_i(t) \qquad (1)$$

for $i = 1, \ldots, N$. The firing rate of each unit, $r_i(t)$, is related to its total input $x_i(t)$ through a rectified $\tanh$ nonlinearity, $r_i(t) = \max(0, \tanh(x_i(t)))$. Every unit in the RNN receives input from all other units through the recurrent weight matrix $W^{\text{rec}}$ and also receives external input, $I(t)$, through the weight matrix $W^{\text{in}}$. These weight matrices are randomly initialized so no structure is a priori introduced into the network. Each unit has an associated bias, $b_i$ which is learned and an associated noise term, $\xi_i(t)$, sampled at every timestep from a Gaussian with zero mean and constant variance. The network was simulated using the Euler method for $T = 500$ timesteps of duration $\tau/10$ ($\tau$ is set to be 250ms throughout the paper).

Let $\theta$ be the current head direction. Input to the RNN is composed of three terms: two inputs encode the initial head direction in the form of $\sin(\theta_0)$ and $\cos(\theta_0)$, and a scalar input encodes both clockwise (CW, negative) and counterclockwise, (CCW, positive) angular velocity at every timestep. The RNN

is connected to two linear readout neurons, $y_1(t)$ and $y_2(t)$, which are trained to track current head direction in the form of $\sin(\theta)$ and $\cos(\theta)$. The activities of $y_1(t)$ and $y_2(t)$ are given by:

$$y_j(t) = \sum_i W_{ji}^{\text{out}} r_i(t) \tag{2}$$

## 2.2 Input statistics

Velocity at every timestep (assumed to be 25 ms) is sampled from a zero-inflated Gaussian distribution (see Fig. 5). Momentum is incorporated for smooth movement trajectories, consistent with the observed animal behavior in flies and rodents. More specifically, we updated the angular velocity as $\text{AV}(t) = \sigma X + \text{momentum} * \text{AV}(t-1)$, where $X$ is a zero mean Gaussian random variable with standard deviation of one. In the Main condition, we set $\sigma = 0.03$ radians/timestep and the momentum to be 0.8, corresponding to a mean absolute AV of $\sim$100 deg/s. These parameters are set to roughly match the angular velocity distribution of the rat and fly (Stackman & Taube, 1998; Sharp et al., 2001; Bender & Dickinson, 2006; Raudies & Hasselmo, 2012). In section 4, we manipulate the magnitude of AV by changing $\sigma$ to see how the trained RNN may solve the integration task differently.

## 2.3 Training

We optimized the network parameters $W^{\text{rec}}$, $W^{\text{in}}$, $b$ and $W^{\text{out}}$ to minimize the mean-squared error in equation (3) between the target head direction and the network outputs generated according to equation (2), plus a metabolic cost for large firing rates ($L_2$ regularization on $r$).

$$E = \sum_{t,j} (y_j(t) - y_j^{\text{target}}(t))^2 \tag{3}$$

Parameters were updated with the Hessian-free algorithm (Martens & Sutskever, 2011). Similar results were also obtained using Adam (Kingma & Ba, 2015).

## 3 Functional and structural properties emerged in the network

We found that the trained network could accurately track the angular velocity (Fig. 1g). We first examined the functional and structural properties of model units in the trained RNN and compared them to the experimental data from the head direction system in rodents and flies.

### 3.1 Emergence of HD cells (compass units) and HD × AV cells (shifters)

**Emergence of different classes of neurons with distinct tuning properties**

We first plotted the neural activity of each unit as a function of HD and AV (Fig. 2a). This revealed two distinct classes of units based on the strength of their HD and AV tuning (see Appendix Fig. 6a,b,c). Units with essentially zero activity are excluded from further analyses. The first class of neurons exhibited HD tuning with minimal AV tuning (Fig. 2f). The second class of neurons were tuned to both HD and AV and can be further subdivided into two populations - one with high firing rate when animal performs CCW rotation (positive AV), the other favoring CW rotation (negative AV) (CW tuned cell shown in Fig. 2g). Moreover, the preferred head direction of each sub-population of neurons tile the complete angular space (Fig. 2b). Embedding the model units into 3D space using t-SNE reveals a clear compass-like structure, with the three classes of units being separated (Fig. 2c).

**Mapping the functional architecture of RNN to neurophysiology**

Neurons with HD tuning but not AV tuning have been widely reported in rodents (Taube et al., 1990a; Blair & Sharp, 1995; Stackman & Taube, 1998), although the HD*AV tuning profiles of neurons are rarely shown (but see Lozano et al. (2017)). By re-analyzing the data from Peyrache et al. (2015), we find that neurons in the anterodorsal thalamic nucleus (ADN) of the rat brain are selectively tuned to HD but not AV (Fig. 2d, also see Lozano et al. (2017)), with HD*AV tuning profile similar to what our model predicts. Preliminary evidence suggests that this might also be true for ellipsoid body (EB) compass neurons in the fruit fly HD system (Green et al., 2017; Turner-Evans et al., 2017).

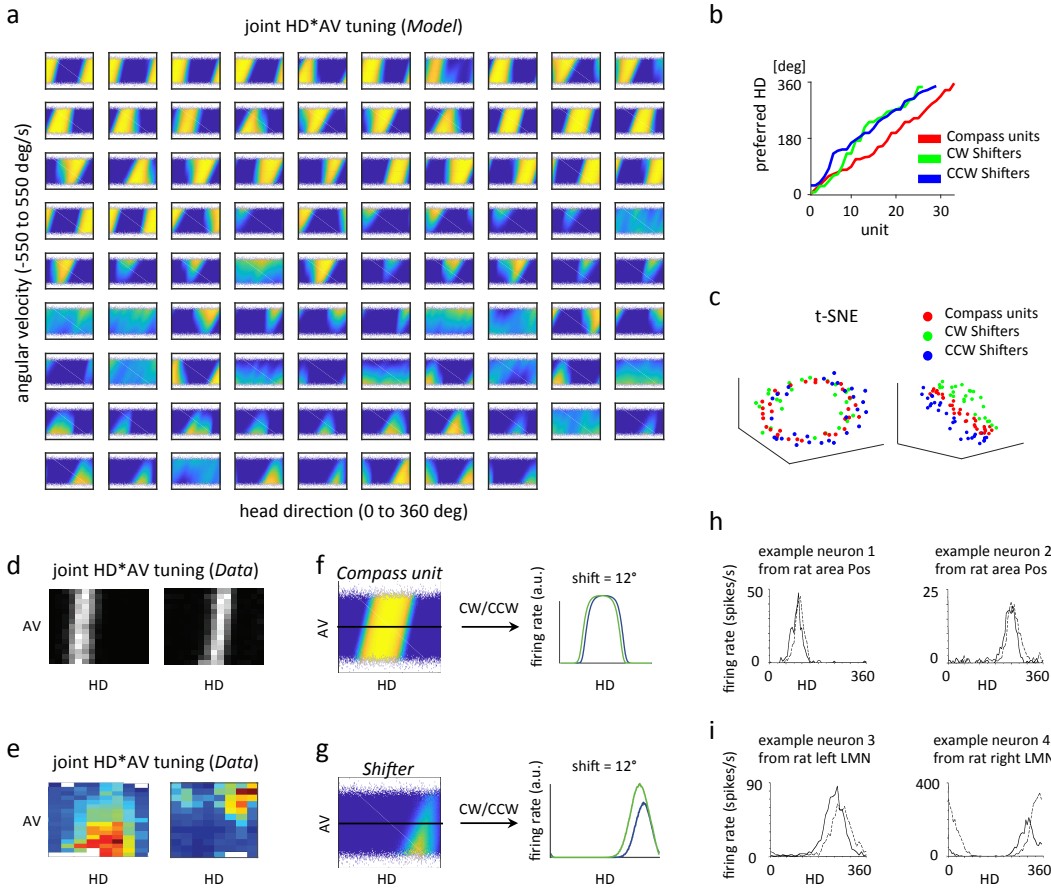

Figure 2: Emergence of different functional cell types in the trained RNN. **a)** Joint HD*AV tuning plots for individual units in the RNN. Units are arranged by functional type (Compass, CCW Shifters, CW Shifters). The activity of each unit is shown as a function of head direction (x-axis) and angular velocity (y-axis). For each unit, AV ranges from -550 to 550 deg/s, and HD ranges from 0 to 360 deg. **b)** Preferred HD of each unit within each functional type. Approximately uniform tiling of preferred HD for each functional type of model neurons is observed. **c)** 3D embedding of model neurons using t-SNE, with the distance between two units defined as one minus their firing rate correlation, exhibits a compass-like structure from one view angle (left) and are segregated according to AV in another view angle (right). Each dot represents one unit. **d)** Joint HD*AV tuning plots for two example HD neurons in the rat anterodorsal thalamic nucleus, or ADN (plotted based on data from Peyrache et al. (2015), downloaded from CRCNS website). White indicates high firing rate. **e)** Joint HD*AV tuning plots for two example neurons from the PB of of the fly central complex, adapted from Turner-Evans et al. (2017). Red indicates high firing rate. **(f,g,h,i)** Detailed tuning properties of model neurons match neural data. **f)** HD tuning curves for model Compass units exhibit shifted peaks at high CW (green) and CCW rotations (blue). **g)** HD tuning curves for model Shifters exhibit peak shift and gain changes when comparing CW (green) and CCW (blue) rotations. **h)** HD tuning curves for CW (solid) and CCW (dashed) conditions for two example neurons in the postsubiculum of rats, adapted from Stackman & Taube (1998). **i)** HD tuning curves for CW (solid) and CCW (dashed) conditions for two example neurons in the lateral mammillary nuclei of rats, adapted from Stackman & Taube (1998).

Neurons tuned to both HD and AV tuning have also been reported previously in rodents and fruit flies (Sharp et al., 2001; Stackman & Taube, 1998; Bassett & Taube, 2001), although the joint HD*AV tuning profiles of neurons have only been documented anecdotally with a few cells (Turner-Evans et al. (2017)). In rodents, certain cells are also observed to display HD and AV tuning (Fig. 2e). In addition, in the fruit fly heading system, neurons on the two sides of the protocerebral bridge (PB) (Pfeiffer & Homberg, 2014) are also tuned to CW and CCW rotation, respectively, and tile the complete angular space, much like what has been observed in our trained network (Green et al., 2017;

Turner-Evans et al., 2017). These observations collectively suggest that neurons that are HD but not AV selective in our model can be tentatively mapped to "Compass" units in the EB, and the two sub-populations of neurons tuned to both HD and AV map to "Shifter" neurons on the left PB and right PB, respectively. We will correspondingly refer to our model neurons as either 'Compass' units or 'CW/CCW Shifters' (Further justification of the terminology will be given in sections 3.2 & 3.3)

**Tuning properties of model neurons match experimental data**

We next sought to examine the tuning properties of both Compass units and Shifters of our network in greater detail. First, we observe that for both Compass units and Shifters, the HD tuning curve varies as a function of AV (see example Compass unit in Fig. 2f and example Shifter in Fig. 2g). Population summary statistics concerning the amount of tuning shift are shown in Appendix Fig. 7a. The preferred HD tuning is biased towards a more CW angle at CW angular velocities, and vice versa for CCW angular velocities. Consistent with this observation, the HD tuning curves in rodents were also dependent upon AV (see example neurons in Fig. 2h,i) (Blair & Sharp, 1995; Stackman & Taube, 1998; Taube & Muller, 1998; Blair et al., 1997; 1998). Second, the AV tuning curves for the Shifters exhibit graded response profiles, consistent with the measured AV tuning curves in flies and rodents (see Fig. 1b,d). Across neurons, the angular velocity tuning curves show substantial diversity (see Appendix Fig. 6b), also consistent with experimental reports (Turner-Evans et al., 2017).

In summary, the majority of units in the trained RNN could be mapped onto the biological head direction system in both general functional architecture and also in detailed tuning properties. Our model unifies a diverse set of experimental observations, suggesting that these neural response properties are the consequence of a network solving an angular integration task optimally.

## 3.2 CONNECTIVITY STRUCTURE OF THE NETWORK

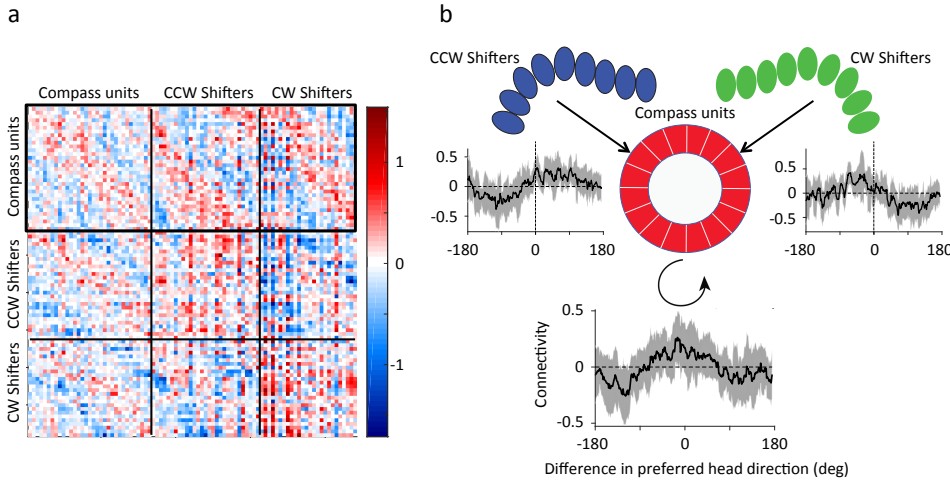

Figure 3: Connectivity of the trained network is structured and exhibits similarities with the connectivity in the fly central complex. **a)** Pixels represent connections from the units in each column to the units in each row. Excitatory connections are in red, and inhibitory connections are in blue. Units are first sorted by functional classes, and then are further sorted by their preferred HD within each class. The black box highlights recurrent connections to the Compass units from Compass units, from CCW Shifters, and from CW Shifters. **b)** Ensemble connectivity from each functional cell type to the Compass units as highlighted in a), in relation to the architecture of the PB & EB in the fly central complex. Plots show the average connectivity (shaded area indicates one s.d.) as a function of the difference between the preferred HD of the cell and the Compass unit it is connecting to. Compass units connect strongly to units with similar HD tuning and inhibit units with dissimilar HD tuning. CCW Shifters connect strongly to Compass units with preferred head directions that are slightly CCW-shifted to its own, and CW Shifters connect strongly to Compass units with preferred head directions that are slightly CW-shifted to its own. Refer to Appendix Fig. 8b for the full set of ensemble connectivity between different classes.

Previous experiments have detailed a subset of connections between EB and PB neurons in the fruit fly. We next analyzed the connectivity of Compass units and Shifters in the trained RNN to ask whether it recapitulates these connectivity patterns - a test which has never been done to our knowledge in any system between artificial and biological neural networks (see Fig. 3).

**Compass units exhibit local excitation and long-range inhibition**

We ordered Compass units, CCW Shifters, and CW Shifters by their preferred head direction tuning and plotted their connection strengths (Fig. 3a). This revealed highly structured connectivity patterns within and between each class of units. We first focused on the connections between individual Compass units and observed a pattern of local excitation and global inhibition. Neurons that have similar preferred head directions are connected through positive weights and neurons whose preferred head directions are anti-phase are connected through negative weights (Fig. 3b). This pattern is consistent with the connectivity patterns inferred in recent work based on detailed calcium imaging and optogenetic perturbation experiments (Kim et al., 2017), with one caveat that the connectivity pattern inferred in this study is based on the effective connectivity rather than anatomical connectivity. We conjecture that Compass units in the trained RNN serve to maintain a stable activity bump in the absence of inputs (see section 3.3), as proposed in previous theoretical models (Turing, 1952; Amari, 1977; Zhang, 1996).

**Asymmetric connectivity from Shifters to Compass units**

We then analyzed the connectivity between Compass units and Shifters. We found that CW Shifters excite Compass units with preferred head directions that are clockwise to its own, and inhibit Compass units with preferred head directions counterclockwise to its own (Fig. 3b). The opposite pattern is observed for CCW Shifters. Such asymmetric connections from Shifters to the Compass units are consistent with the connectivity pattern observed between the PB and the EB in the fruit fly central complex (Lin et al., 2013; Green et al., 2017; Turner-Evans et al., 2017), and also in agreement with previously proposed mechanisms of angular integration (Skaggs et al., 1995; Green et al., 2017; Turner-Evans et al., 2017; Zhang, 1996) (Fig. 3b). We note that while the connectivity between PB Shifters and EB Compass units are one-to-one (Lin et al., 2013; Wolff et al., 2015; Green et al., 2017), the connectivity profile in our model is broad, with a single CW Shifter exciting multiple Compass units with preferred HDs that are clockwise to its own, and vice versa for CCW Shifters.

In summary, the RNN developed several anatomical features that are consistent with structures reported or hypothesized in previous experimental results. A few novel predictions are worth mentioning. First, in our model the connectivity between CW and CCW Shifters exhibit specific recurrent connectivity (Fig. 8). Second, the connections from Shifters to Compass units exhibit not only excitation in the direction of heading motion, but also inhibition that is lagging in the opposite direction. This inhibitory connection has not been observed in experiments yet but may facilitate the rotation of the neural bump in the Compass units during turning (Wolff et al., 2015; Franconville et al., 2018; Green et al., 2017; Green & Maimon, 2018). In the future, EM reconstructions together with functional imaging and optogenetics should allow direct tests of these predictions.

## 3.3 PROBING THE COMPUTATION IN THE NETWORK

We have segregated neurons into Compass and Shifter populations according to their HD and AV tuning, and have shown that they exhibit different connectivity patterns that are suggestive of different functions. Compass units putatively maintain the current heading direction and Shifter units putatively rotate activity on the compass according to the direction of angular velocity. To substantiate these functional properties, we performed a series of perturbation experiments by lesioning specific subsets of connections.

**Perturbation while holding a constant head direction**

We first lesioned connections when there is zero angular velocity input. Normally, the network maintains a stable bump of activity within each class of neurons, *i.e.,* Compass units, CW Shifters, and CCW Shifters (see Fig. 4a,b). We first lesioned connections from Compass units to all units and found that the activity bumps in all three classes disappeared and were replaced by diffuse activity in a large proportion of units. As a consequence, the network could not report an accurate estimate of its current heading direction. Furthermore, when the connections were restored, a bump formed

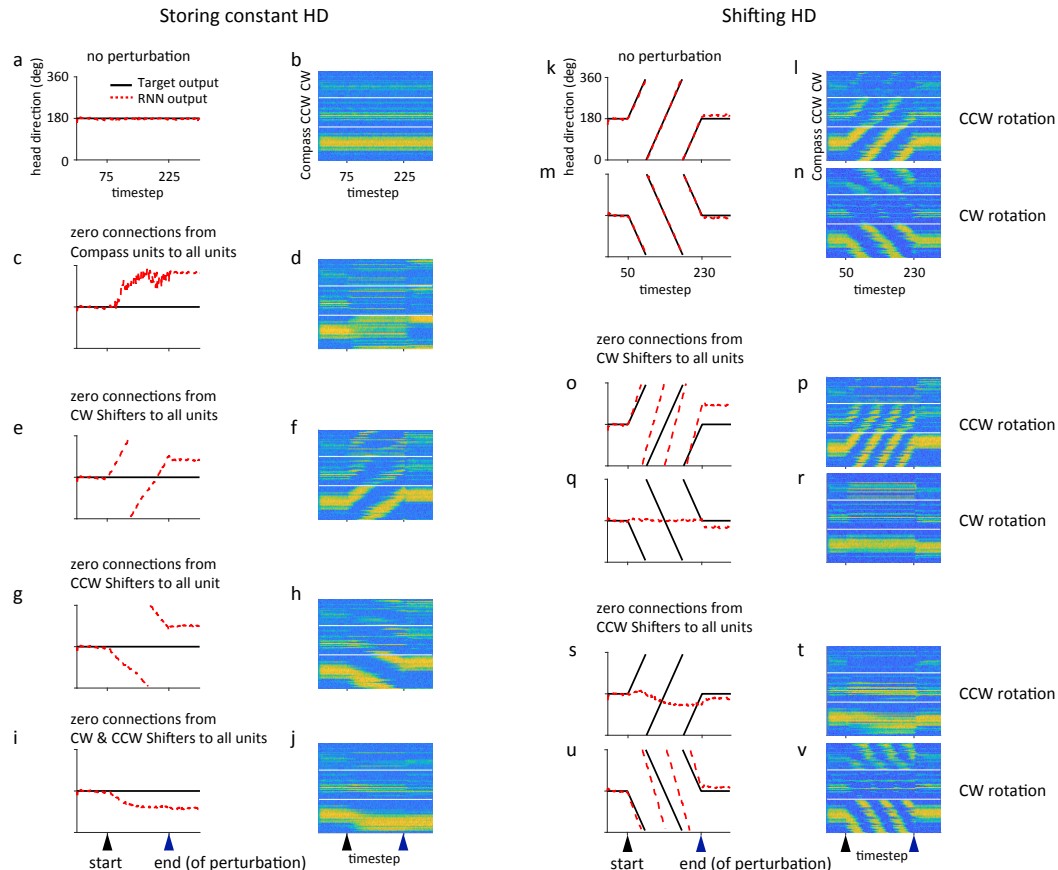

Figure 4: Probing the functional role of different classes of model neurons. **a-j)** Perturbation analysis in the case of maintaining a constant HD. **a)** Under normal conditions, the RNN output matches the target output. **b)** Population activity for the trial shown in a), sorted by the preferred HD and the class of each unit, i.e., Compass units, CCW Shifters, CW Shifters. **c-j)** RNN output and population activity when a specific set of connections are set to zero during the period indicated by blue arrows in i) and j). **k-v)** Perturbation analysis in the case of a shifting HD. For each manipulation, CW rotation and CCW rotation are tested, resulting in two trials. **k-n)** Normal case without any perturbation. **o-v)** RNN output and population activity when connections from CW Shifters (o-r) and CCW Shifters (s-v) are set to zero. Refer to the main text for the interpretation of the results.

again without any external input (Fig. 4d), suggesting the network can spontaneously generate an activity bump through recurrent connections mediated by Compass units.

We then lesioned connections from CW Shifters to all units and found that all three bumps exhibit a CCW rotation, and the read-out units correspondingly reported a CCW rotation of heading direction (Fig. 4e,f). Analogous results were obtained with lesions of CCW Shifters, which resulted in a CW drifting bump of activity (Fig. 4g,h). These results are consistent with the hypothesis that CW and CCW Shifters simultaneously activate the compass, with mutually cancelling signals, even when the heading direction is stationary. When connections are lesioned from both CW and CCW Shifters to all units, we observe that Compass units are still capable of holding a stable HD activity bump (Fig. 4i,j), consistent with the predictions that while CW/CCW Shifters are necessary for updating heading during motion, Compass units are responsible for maintaining heading.

**Perturbation while integrating constant angular velocity**

We next lesioned connections during either constant CW or CCW angular velocity. Normally, the network can integrate AV accurately (Fig. 4k-n). As expected, during CCW rotation, we observe a corresponding rotation of the activity bump in Compass units and in CCW Shifters, but CW Shifters display low levels of activity. The converse is true during CW rotation. We first lesioned connections from CW Shifters to all units, and found that it significantly impaired rotation in the CW direction,

and also increased the rotation speed in the CCW direction. Lesioning of CCW Shifters to all units had the opposite effect, significantly impairing rotation in the CCW direction. These results are consistent with the hypothesis that CW/CCW Shifters are responsible for shifting the bump in a CW and CCW direction, respectively, and are consistent with the data in Green et al. (2017), which shows that inhibition of Shifter units in the PB of the fruit fly heading system impairs the integration of HD. Our lesion experiments further support the segregation of units into modular components that function to separately maintain and update heading during angular motion.

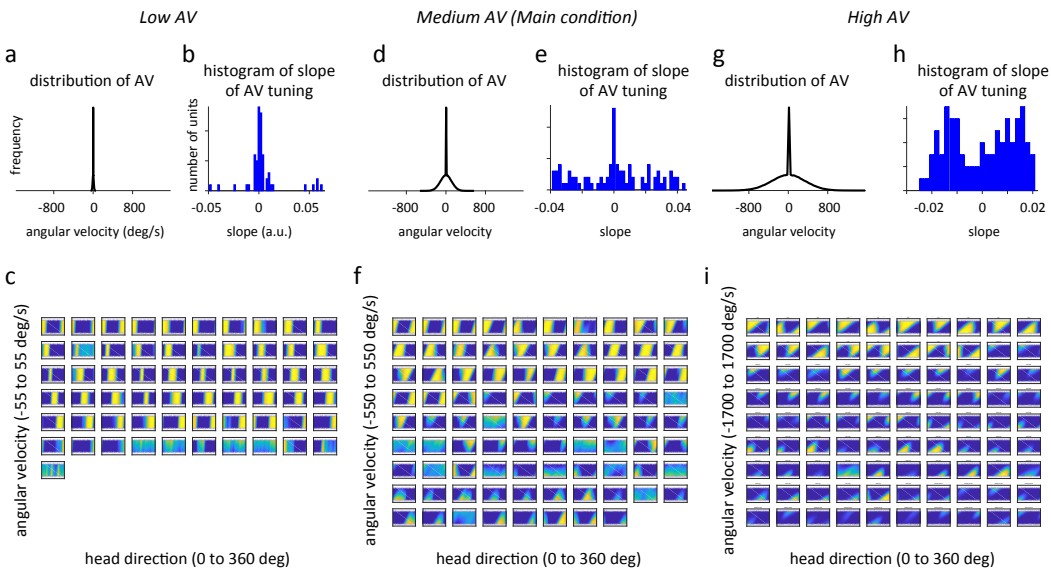

Figure 5: Representations in the trained RNN vary as the input statistics change. **a)** The AV distribution used to train the RNN in the low angular velocity condition. **b)** A histogram of the slopes of the AV tuning curves for individual units in the low AV condition. **c)** Heatmaps of the joint AV and HD tuning for each unit in the low AV condition. Units are arranged by functional type (Compass, CCW Shifters, CW Shifters). The activity of each unit is shown as a function of head direction (x-axis) and angular velocity (y-axis). **d,e,f)** Same convention as a-c, but for the main condition (f is the same as Fig. 2a). **g,h,i)** Same convention as a-c, but for the high angular velocity condition.

## 4   ADAPTATION OF NETWORK PROPERTIES TO INPUT STATISTICS

Optimal computation requires the system to adapt to the statistical structure of the inputs (Barlow, 1961; Attneave, 1954). In order to understand how the statistical properties of the input trajectories affect how a network solves the task, we trained RNNs to integrate inputs generated from low and high AV distributions.

When networks are trained with small angular velocities, we observe the presence of more units with strong head direction tuning but minimal angular velocity tuning. Conversely, when networks are trained with large AV inputs, fewer Compass units emerge and more units become Shifter-like and exhibit both HD and AV tuning (Fig. 5c,f,i). We sought to quantify the overall AV tuning under each velocity regime by computing the slope of each neuron's AV tuning curve at its preferred HD angle. We found that by increasing the magnitude of AV inputs, more neurons developed strong AV tuning (Fig. 5b,e,h). In summary, with a slowly changing head direction trajectory, it is advantageous to allocate more resources to hold a stable activity bump, and this requires more Compass units. In contrast, with quickly changing inputs, the system must rapidly update the activity bump to integrate head direction, requiring more Shifter units. This prediction may be relevant for understanding the diversity of the HD systems across different animal species, as different species exhibit different overall head turning behavior depending on the ecological demand (Stone et al., 2017; Seelig & Jayaraman, 2015; Heinze, 2017; Finkelstein et al., 2018).

## 5 DISCUSSION

Previous work in the sensory systems have mainly focused on obtaining an optimal representation (Barlow, 1961; Laughlin, 1981; Linsker, 1988; Olshausen & Field, 1996; Simoncelli & Olshausen, 2001; Yamins et al., 2014; Khaligh-Razavi & Kriegeskorte, 2014) with feedforward models. Several recent studies have probed the importance of recurrent connections in understanding neural computation by training RNNs to perform tasks (*e.g.,* Mante et al. (2013); Sussillo et al. (2015); Cueva & Wei (2018)), but the relation of these trained networks to the anatomy and function of brain circuits are not mapped. Using the head direction system, we demonstrate that goal-driven optimization of recurrent neural networks can be used to understand the functional, structural and mechanistic properties of neural circuits. While we have mainly used perturbation analysis to reveal the dynamics of the trained RNN, other methods could also be applied to analyze the network. For example, in Appendix Fig. 10, using fixed point analysis (Sussillo & Barak, 2013; Maheswaranathan et al., 2019), we found evidence consistent with attractor dynamics. Due to the limited amount of experimental data available, comparisons regarding tuning properties and connectivity are largely qualitative. In the future, studies of the relevant brain areas using Neuropixel probes (Jun et al., 2017) and calcium imaging (Denk et al., 1990) will provide a more in-depth characterization of the properties of HD circuits, and will facilitate a more quantitative comparison between model and experiment.

In the current work, we did not impose any additional structural constraint on the RNNs during training, asides from prohibiting self-connections. We have chosen to do so in order to see what structural properties would emerge as a consequence of optimizing the network to solve the task. It is interesting to consider how additional structural constraints affect the representation and computation in the trained RNNs. One possibility would to be to have the input or output units only connect to a subset of the RNN units. Another possibility would be to freeze a subset of connections during training. Future work should systematically explore these issues.

Recent work suggests it is possible to obtain tuning properties in RNNs with random connections (Sederberg & Nemenman, 2019). We found that training was necessary for the joint HD*AV tuning (see Appendix Fig. 9) to emerge. While Sederberg & Nemenman (2019) consider a simple binary classification task, our integration task is computationally more complicated. Stable HD tuning requires the system to keep track of HD by accurate integration of AV, and to stably store these values over time. This computation might be difficult for a random network to perform and, more generally, completely random networks may lead to different memory representations than the attractor geometry we observe in Fig. 10 (Cueva et al., 2019).

Our approach contrasts with previous network models for the HD system, which are based on hand-crafted connectivity (Zhang, 1996; Skaggs et al., 1995; Xie et al., 2002; Green et al., 2017; Kim et al., 2017; Knierim & Zhang, 2012; Song & Wang, 2005; Kakaria & de Bivort, 2017; Stone et al., 2017). Our modeling approach optimizes for task performance through stochastic gradient descent. We found that different input statistics lead to different heading representations in an RNN, suggesting that the optimal architecture of a neural network varies depending on the task demand - an insight that would be difficult to obtain using the traditional approach of hand-crafting network solutions. Although we have focused on a simple integration task, this framework should be of general relevance to other neural systems as well, providing a new approach to understand neural computation at multiple levels.

Our model may be used as a building block for AI systems to perform general navigation (Pei et al., 2019). In order to effectively navigate in complex environments, the agent would need to construct a cognitive map of the surrounding environment and update its own position during motion. A circuit that performs heading integration will likely be combined with another circuit to integrate the magnitude of motion (speed) to perform dead reckoning. Training RNNs to perform more challenging navigation tasks such as these, along with multiple sources of inputs, i.e., vestibular, visual, auditory, will be useful for building robust navigational systems and for improving our understanding of the computational mechanisms of navigation in the brain (Cueva & Wei, 2018; Banino et al., 2018).

### ACKNOWLEDGMENTS

Research supported by NSF NeuroNex Award DBI-1707398 and the Gatsby Charitable Foundation. We would like to thank Kenneth Kay for careful reading of an earlier version of the paper, and Rong Zhu for help preparing panel d in Figure 2.

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

# A APPENDIX

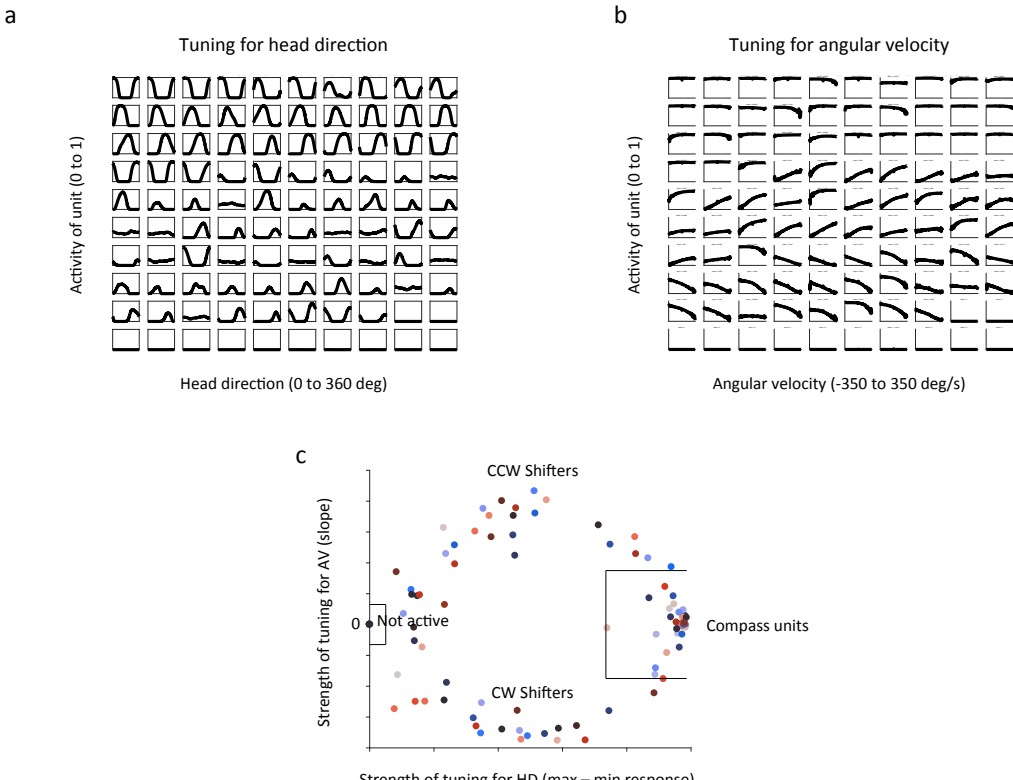

Figure 6: Tuning properties and unit classification. **a)** HD tuning curves for all 100 units in the RNN based on the Main condition. Units are arranged by functional type (Compass, CCW Shifters, CW Shifters). The activity of each unit is shown as a function of head direction. The 12 units at the bottom were not active, did not contribute to the network, and were not included in the analyses. All 88 units with some activity were included in the analyses. **b)** Similar to a), but for AV tuning curves. **c)** Classification into different populations using HD and AV tuning strength.

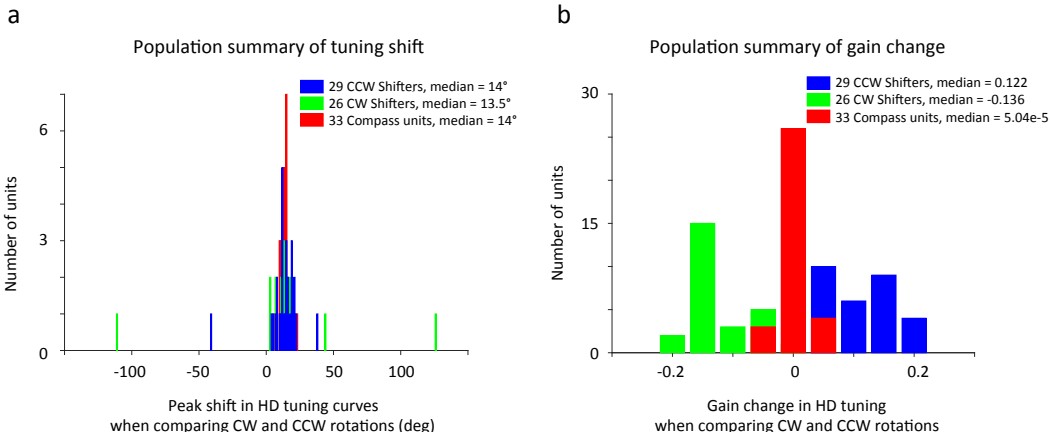

Figure 7: Population summary of tuning shift and gain change when comparing the CW and CCW rotations. **a)** Population summary of tuning shift. **b)** Population summary of change of the peak firing rate of HD tuning curves.

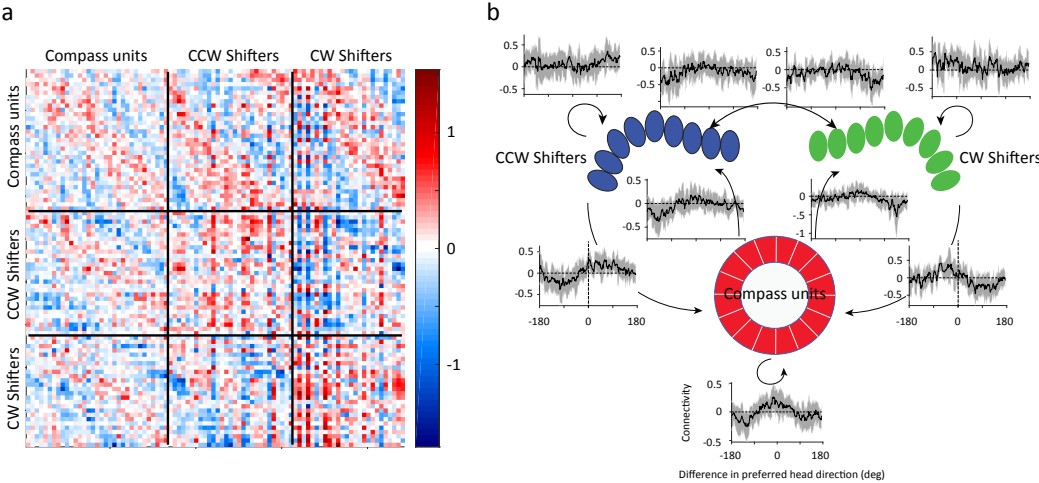

Figure 8: Connectivity of the trained network. **a)** Pixels represent connections from the units in each column to the units in each row. Units are first sorted by functional classes, and then are further sorted by their preferred HD within each class. Excitatory connections are in red, and inhibitory connections are in blue. **b)** Ensemble connectivity from each functional cell type. Plots show the average connectivity (shaded area indicates one standard deviation) as a function of the difference in preferred HD.

a                                                              b

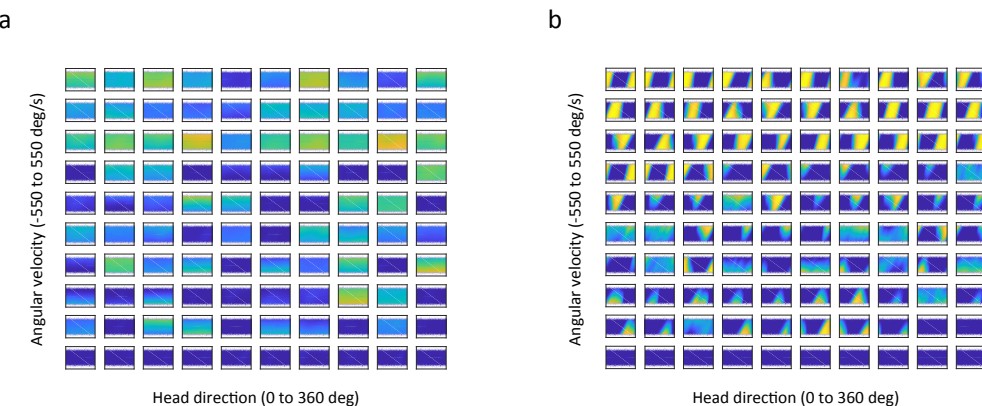

Figure 9: Joint HD × AV tuning of the initial, randomly connected network and the final trained network. **a)** Before training, the 100 units in the network do not have pronounced joint HD × AV tuning. The color scale is different for each unit (blue = minimum activity, yellow = maximum activity) to maximally highlight any potential variation in the untrained network. **b)** After training, the units are tuned to HD × AV, with the exception of 12 units (shown at the bottom) which are not active and do not influence the network.

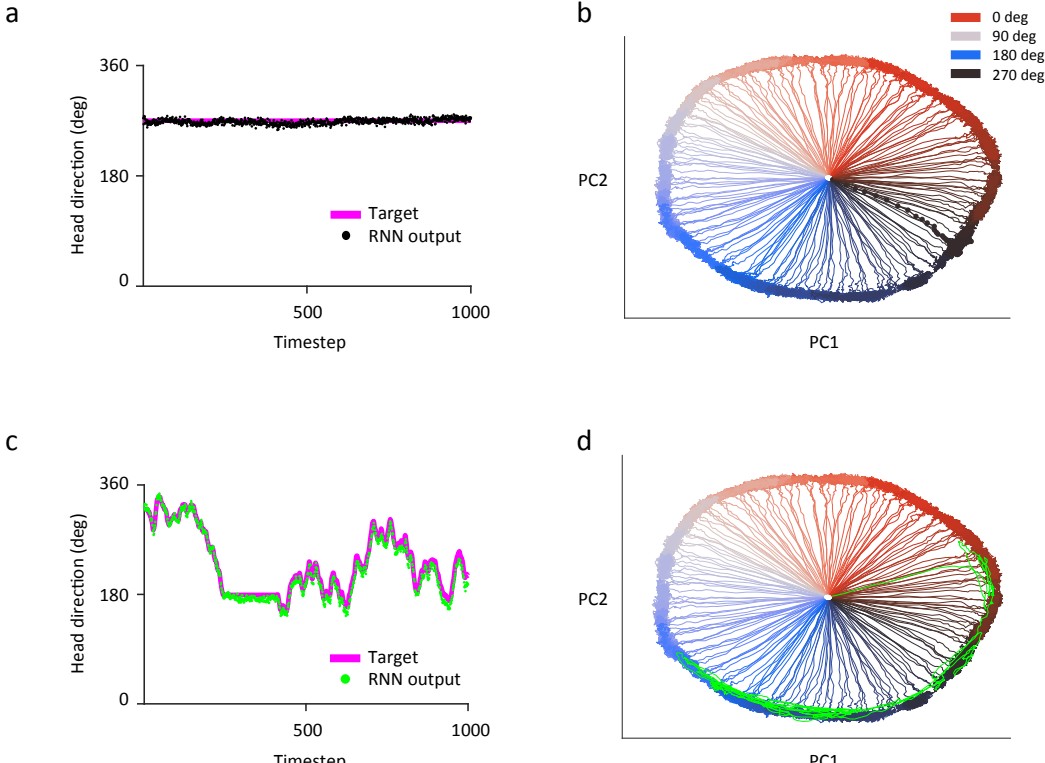

Figure 10: Attractor structure of the trained network. To store angular information the neural activity settles into a compass-like geometry, with the position along the compass determined by the angle. **a)** The RNN is able to store angular information in the presence of noise. In this example sequence an initial input to the RNN specifies the initial heading direction as 270 degrees. The angular velocity input to the RNN is zero and so the network must continue to store this value of 270 degrees throughout the sequence. **b)** The activity of all units in the RNN are shown for 180 sequences, similar to a), after projecting onto the two principal components capturing most of the variance (91%). Each line shows the neural trajectory over time for a single sequence. For example, the neural activity that produces the output shown in a) is highlighted with black dots (1 dot for each timestep). The activity of the RNN is initialized to be the same for all sequences, i.e. $x(0)$ in equation 1 is the same, so the activity for all sequences starts at the same location, namely, the origin of the figure. The activity quickly settles into a compass-like geometry, with the location around the compass determined by the heading direction. **(c,d)** When the RNN is integrating a nonzero angular velocity it appears to transition between the attractors shown in b), i.e. the attractors found when the network was only storing a constant angle. Panel c) shows an example sequence with a nonzero angular velocity. In d) the neural trajectory from c) is superimposed over the attractor geometry from b).

