# OpenReview forum: "Emergence of functional and structural properties of the head direction system by optimization of recurrent neural networks"
_ICLR.cc/2020/Conference — Accept (Spotlight)_

### Official Review · AnonReviewer2 · 2019-10-23
**Official Blind Review #2**

**Rating:** 6

**Review:**

The authors train RNNs to integrate motion cues, to determine the head direction of a simulated animal. They then investigate the tuning properties of the units in the RNN, and the connectivity between units, and make comparisons to the corresponding systems in the fly and rodent.

Similarities are quite strong: the RNN learns to reproduce the ring attractor seen in the fly, where head direction (HD) cells excite other HD cells with similar direction preference and inhibit those with dissimilar preference. Shifter cells are also observed, that integrate angular velocity cues and "shift" the representation of head direction: these show an intuitive asymmetric connectivity profile.

I can't quite decide what to make of this paper. One the one hand, the recapitulation of the biological circuit in the RNN is fairly compelling. On the other hand, other models (admittedly with hand-tuned connectivity) yield quite similar findings. So my overall opinion comes down to the question of: how big a deal is it that the RNN learned this connectivity, instead of it being designed by a modeller like Xiao-Jing Wang?

I have a few specific suggestions:

1) It would be useful as a comparison to duplicate the analysis of unit tuning, connectivity, etc., in untrained networks. Recent work (arXiv:1909.10116 [q-bio.NC]) shows that, even in untrained randomly-connected RNNs, some units have (by chance) tuning, and that tuning depends on the connectivity between units in a manner that could somewhat resemble the attractor models.

2) There were important details that I couldn't find. For example, how many of the RNN units receive the external inputs? Is it all of them? Is that realistic?

3) I question the approach (first paragraph of Sec. 3.1) of only analyzing the units  with strong HD and/or AV selectivity. Presumably the mouse shows some units with weak tuning, and investigating the connectivity of the weakly-tuned units in the RNN (e.g., their connectivity to each other, and to HD and AV units) could be informative.

One analysis to consider: ablate the nonselective units after training, and run the RNN with no retraining. Does the head direction tracking function remain intact or degrade? Based on some previous work with feedforward nets (arXiv:1803.06959 [stat.ML]), I'd guess that the function could be seriously degraded by removing nonselective units. That modelling result could make a strong prediction for experiments: inactivation of nonselective neurons (e.g., using archaerhodopsin), would impair function.



**Experience Assessment:**

I have published in this field for several years.

**Review Assessment: Checking Correctness Of Derivations And Theory:**

N/A

**Review Assessment: Checking Correctness Of Experiments:**

I assessed the sensibility of the experiments.

**Review Assessment: Thoroughness In Paper Reading:**

I read the paper thoroughly.

---

> ### Author Response · Authors · 2019-11-15
> **Response to reviewer 2**
>
> Thank you for your positive assessment and suggestions. We address your concerns/suggestions below.
>
> 1. Difference between our approach and the traditional approach using hand-crafted connectivity
>
> The reviewer touched on an important and more general point, namely the value of using optimization-based RNNs as opposed to models with handcrafted connectivity, which have been the tradition in computational neuroscience.  Although both approaches may lead to networks that could solve a task reasonably well, the interpretation of the two approaches can be quite different. Our approach suggests that the network architecture that has emerged from stochastic gradient descent is advantageous in performing angular integration relative to all other proposed model architectures, of which there are many (Zhang, 1996; Skaggs et al., 1995; Xie et al., 2002; Green et al., 2017; Kim et al., 2017; Knierim & Zhang, 2012; Song & Wang, 2005; Kakaria & de Bivort, 2017; Stone et al., 2017). This interpretation is typically difficult to make using networks with hand-crafted activity. The result that trained networks in the Main condition share some similarities to previous hand-crafted network models for the HD system is, to us, surprising and interesting. Furthermore, we show that different representations could emerge as the optimal strategy to solve the task in different regimes. This is a novel insight which would be difficult to claim with the traditional hand-crafted approach.
>
> These points, however, were at best implicit in the initial version. We now discuss these points explicitly in the Discussion. We thank the reviewer for bringing up this important point, and have done our best to address it.
>
> In addition to learning about this particular circuit, we also had a more general motivation for this work. We wanted to know if artificial neural networks can predict not only neural activity, as has been shown previously, but if the architecture is unconstrained and optimized, the anatomical properties of neural circuits. This work suggests that artificial neural networks can be used, at least in some cases, to study the brain at the level of both neural activity and anatomical organization.
>
> 2. Randomly connected network
>
> Following the reviewer’s suggestions, we have checked the tuning properties in an untrained randomly initialized network. We have now included a new figure in the Appendix Fig. 9. From this plot, it is evident that the joint HD*AV tuning properties developed only after training. In the paper (arXiv:1909.10116 [q-bio.NC]) cited by the reviewer, the RNN is performing a simple binary classification task, while our task is more complicated, requiring the network to properly integrate the inputs. We speculate that the difference in the difficulty of the task may be responsible for the discrepancy. In addition there is some evidence that randomly connected networks are not able to robustly store information over time (Cueva et al. bioRxiv doi: 10.1101/504936), a crucial ability for our task. We think this is an interesting point and have added a paragraph in the Discussion to discuss this.
>
> 3. Connectivity between inputs and RNN
>
> In our model, all of the units in the RNN (N=100) received external inputs. As this is the first attempt to study the HD system through the training of an RNN, we chose to not impose structural constraints, such that we can observe the structural features that emerged as a consequence of task optimization. We have now clarified this by stating that "Every unit in the RNN receives input from all other units through the recurrent weight matrix $W^{\mathrm{rec}}$ and also receives external input, $I(t)$, through the weight matrix $W^{\mathrm{in}}$”.
>
> Alternatively, as the reviewer hinted, it might be reasonable to assume some additional constraints on the network connectivity. Following this line of thinking, one could start to train by adding more architectural constraint. We have indeed thought about this. Although we didn’t pursue this direction in the current paper, this would be a very interesting direction for the future.  We have now added a discussion of this point in the third paragraph of the Discussion section.
>
> 4. Selection of the RNN units
>
> We agree with the reviewer (note that Reviewer 1 raised a similar concern). Indeed, it is entirely possible that units with minimal tuning could be crucial for the computation in the network. We initially excluded 15 units (12 "dead units” with close to zero activity, plus 3 weakly tuned units) in the analyses. Following the reviewer's suggestions, now we have relaxed our criterion. We have included the 3 weakly tuned units and re-done the analyses. The initial results still hold. For the "dead units”, we verified that lesioning the 12 "dead units” has a minimal effect on the network's performance (far below what can be seen visibly) on tracking the HD as well as influencing the dynamics of other units in the RNN.

---

### Official Review · AnonReviewer3 · 2019-10-24
**Official Blind Review #3**

**Rating:** 8

**Review:**

This paper examines head direction representations in a RNN. The RNN was trained to report current head direction using initial head direction and angular velocity information as inputs. The authors compare the representations in the RNN to the representations found in the head direction systems of mice and flies. They find that the representations and connectivity matrices of the RNN recapitulate many aspects of the real brains’ head direction systems. These include the presence of cells tuned only for head direction (“ring cells”) and cells with combined head direction and angular velocity tuning (“shift cells”). These cells tile the space in a functional torus structure, as revealed by t-SNE, which resembles the structures seen in real brains. Similarly, the connectivity profiles and functional role of these cell types matches what is observed in the brains. Altogether, these results demonstrate that RNNs optimized for this task can capture both the functional and anatomical aspects of the systems found in animals. This validates the use of RNNs for studying these systems. In my opinion, it also suggests that information about the structure of brains could inform strong priors for ANNs.

I think this is a great paper. It was a pleasure to read, the data was very clear, and the results very interesting. It should be accepted in my opinion. I have only a few minor notes for improvement.

- Per my last note in the summary, it would be nice if the authors made the paper a little bit more relevant for machine learners by discussing how this information could potentially inform novel neural network architectures for navigation.

- There’s a missing reference on page 3, second last paragraph.

- Figure 5 a,d, and g would be easier to interpret if they all had the same x-range as g (that way, you could see the narrower ranges and wouldn’t have to notice that the scales were different).

- Why have Figure S8 and Fig 3 separate? Aren’t they nearly identical? Fig S8 isn’t that much bigger, why not just make it Fig. 3?

**Experience Assessment:**

I have published in this field for several years.

**Review Assessment: Checking Correctness Of Derivations And Theory:**

N/A

**Review Assessment: Checking Correctness Of Experiments:**

I carefully checked the experiments.

**Review Assessment: Thoroughness In Paper Reading:**

I read the paper thoroughly.

---

> ### Author Response · Authors · 2019-11-15
> **Response to reviewer 3**
>
> Thank you for your positive evaluation of our paper and suggestions. You mention that it would be useful to make connections to the machine learners. We agree. We have now added a paragraph in the Discussion. In particular, we discuss how one may be able to use our model as a building block for larger-scale navigation systems. We also fixed a few minor issues according to your suggestions, including making the axes in Fig. 5 consistent. For Fig. 3, we tried to replace it with the more complete version (i.e., Fig. 8). But we feel that it is a bit too crowded and there is an overwhelming amount of information, which might distract the reader. Thus we have decided to keep the current Fig. 3. However, we believe the full set of connectivity motifs shown in Fig. 8 are interesting and may be important for connecting this work to future experiments and so are worth keeping in the Appendix.

---

### Official Review · AnonReviewer1 · 2019-10-24
**Official Blind Review #1**

**Rating:** 6

**Review:**

## Overview
This paper studies whether a recurrent neural network trained to solve a particular task (integration of angular velocity to generate head direction). Given trained recurrent networks that solve the task, the paper analyzes these using a number of different methods (visualizing tuning curves, perturbation experiments, and varying input statistics). Overall, I think this is a nice paper that shows the power of using artificial networks as a model system to answer neuroscientifically motivated questions. In particular, I found the perturbation analyses particularly illuminating. These kinds of experiments are only possible in these artificial networks, and can highlight/guide future biological experiments.

## Major comments
- "Units with minimal tuning to both variables are discarded from further analysis"  -- how many units end up being discarded? The reason I ask is because the number of discarded neurons affects this statement: "Therefore, units in the trained RNN could be mapped on to the biological head direction system both in terms of general functional architecture and detailed tuning properties.". If a large number of neurons are discarded, then the statement should be amended to say that "a fraction of units in the trained RNN" can be mapped onto the HD system. However, my understanding is that many biological neurons have tuning properties that are hard to classify, perhaps the discarded neurons could map on to these previously uncharacterized neurons?
- The paper demonstrates a number of qualitiative similarities between artificial and biological networks (e.g. both contain neurons tuned for HD or AV). It would be even more compelling if, wherever possible, these comparisons were made to be quantitative.
- "We conjecture that Ring units in the trained RNN serve to maintain a stable activity bump in the absence of inputs". I think a cool direct test of this idea would be to use the techniques in Barak & Sussillo 2013 (find fixed points of the recurrent network dynamics, and analyze the linearized system at those fixed points) to identify the ring attractor structure. In particular, numerical auto-differentiation can be used to automatically identify these points (c.f. the methods in https://arxiv.org/abs/1907.08549). Using these tools, can one find the ring attractor hidden in these networks?

## Minor comments/questions
- The layout in Fig 2a (and Fig 5c, 5f, and 5i) is a little misleading. If I understand correctly, the axes labels apply to each individual panel (which shows tuning for a particular neuron). However, since the labels extend across many panels, it looks as if the panels themselves are organized according to angular velocity and head direction, which doesn't make sense.
- Missing citation towards the end of pg. 3
- Consider setting the panels in Fig 5a-h to have the same axes limits, for easier comparison.

**Experience Assessment:**

I have published one or two papers in this area.

**Review Assessment: Checking Correctness Of Derivations And Theory:**

N/A

**Review Assessment: Checking Correctness Of Experiments:**

I assessed the sensibility of the experiments.

**Review Assessment: Thoroughness In Paper Reading:**

I read the paper thoroughly.

---

> ### Author Response · Authors · 2019-11-15
> **Response to reviewer 1**
>
> We thank the reviewer for the positive evaluation of our paper and the constructive feedback. We address your concerns/suggestions below:
>
> 1. Exclusion of model units
>
> In the initial submission, 3 units were weakly tuned and not analyzed in the Main condition (Medium AV), 1 unit was weakly tuned and not analyzed in the Low AV condition, and 0 units were weakly tuned in the High AV condition. In the revision, we have included all units with weak tuning in all of the analyses by assigning them to their closest functional type (Ring, CW shifter, CCW shifter).  We thought it is fair to say that this is a relatively small portion of units in the network, and have changed our statement to "the majority of the units in the trained RNN could be mapped onto the biological head direction system… ”.   There were also "dead units” that showed essentially no activity. For example, see the 12 units (out of 100) at the bottom of Figures 6a and 6b. These units did not contribute to the network as we verified by exactly zeroing their activity and they were not included in the analyses. A subset of neurons often are not active in our angular integration task. This likely could be changed by using a different training procedure, or removing the regularization encouraging low firing rate. Due to the scope of this paper, we did not pursue this further.
>
>
> 2. Comparison to the data
>
> We fully agree that a more quantitative comparison between the model and experimental data would make the paper even stronger. Concerning the model, we did several quantifications in Fig. 2b and Appendix Fig. 7. However, the main hurdle currently comes from a dearth of experimental data. So far, available data is mainly derived from single cell electrophysiology, and recordings are therefore subject to heavily biased sampling. Therefore, it is possible that many of the neurons in the same area where the HD neurons were recorded do not have strong HD tuning or AV tuning, but we cannot claim that is indeed the case based on existing neural data. In the future, NeuroPixel or calcium imaging of the relevant brain areas would provide a more unbiased characterization of the properties of HD circuits, which would facilitate a more quantitative comparison. We have added a sentence in the discussion to reflect these points, and we thank the reviewer for this set of suggestions.
>
> 3. Fixed point dynamics
>
> We think this is a very good idea. In the revised manuscript, we have added a figure in the Appendix (Fig. 10) exploring the fixed points of the network. When the network is simulated with no noise the network settles into fixed points to store heading direction (technically, the magnitude of dx/dt is very small but nonzero as is also the case in the work of Barak & Sussillo 2013). When the network is simulated with noise (as shown in Fig. 10) the neural trajectories jitter around these fixed points. We thank the reviewer for this helpful suggestion.
>
> Minor issues:
> -	Layout of Fig. 2a (and Fig. 5c, 5f, and 5i). We have added sentences to the caption of Fig. 2a to clarify the scheme of the labeling to help avoid confusion. We tried several layouts and found this is one of the better ones. Labelling individual panels would result in a very crowded figure. We have thus decided to keep the layout in the figure.
> -	We replotted panels in Fig. 5a-h to standardize axis limits, according to the reviewer’s suggestion.

---

### Author Response · Authors · 2019-11-15
**Overall statement to all three reviewers**

We would like to thank all three reviewers for carefully reading our paper and for their positive evaluations and constructive feedback. We have run additional analyses according to the reviewers’ suggestions, and we believe that we have addressed or clarified all the concerns raised. We have uploaded a new version which reflects these changes. None of these new analyses have changed any of the main results that were reported in the initial submission. We thank the reviewers’ for their suggestions to improve the paper, and we address each reviewer individually below.

---

### Decision · Program_Chairs · 2019-12-19

**Decision:**

Accept (Spotlight)

**Comment:**

This paper studies properties that emerge in an RNN trained to report head direction, showing that several properties in natural neural circuits performing that function are detected.
All reviewers agree that this is quite an interesting paper. While there are some reservations as to the value of letting a property of interest emerge as opposed to simply hand-coding it in, this approach is seen as powerful and valuable by many people, in that it suggests a higher plausibility that the emerging properties are actually useful when optimizing for that function -- a claim which hand-coding would not make possible. Reviewers have also provided valuable suggestions and requests for clarifications, and authors have responded by improving the presentation and providing more insights.
Overall, this is a solid contribution that will be of interest to the part of the ICLR audience that is interested in biological systems.